# Reevaluating Theoretical Analysis Methods for Optimization in Deep Learning

## Abstract

There is a significant gap between our theoretical understanding of optimization algorithms used in deep learning and their practical performance. Theoretical development usually focuses on proving convergence guarantees under a variety of different assumptions, which are themselves often chosen based on a rough combination of intuitive match to practice and analytical convenience. In this paper, we carefully measure the degree to which the standard optimization analyses are capable of explaining modern algorithms. To do this, we develop new empirical metrics that compare real optimization behavior with analytically predicted behavior. Our investigation is notable for its tight integration with modern optimization analysis: rather than simply checking high-level assumptions made in the analysis (e.g. smoothness), we also verify key low-level identities used by the analysis to explain optimization behavior that might hold even if the high-level motivating assumptions do not. In general, we find that real optimizers often make progress even when typical optimization analysis suggests that they should not. This highlights a need for developing new theoretical frameworks that are better aligned with practice.

## 1 Introduction

In optimization theory, algorithmic development and analysis requires a set of assumptions about the functions we aim to optimize. These assumptions fundamentally influence the behavior of optimization algorithms and their efficacy in practice. For example, Adagrad (Duchi et al., 2011; McMahan & Streeter, 2010) (which later inspired Adam (Kingma & Ba, 2014)) classically relies on the convexity assumption to provide a theoretical convergence guarantee. When the loss is non-convex, a variety of alternate assumptions are deployed, such as smoothness (e.g. a bounded Hessian) (Ghadimi & Lan, 2013; Carmon et al., 2017; Li & Orabona, 2019; Ward et al., 2020; Wang et al., 2023) or "weak convexity" (Davis & Drusvyatskiy, 2019; Mai & Johansson, 2020; Liu et al., 2023b). If these conditions are not met, the convergence analyses of these algorithms may longer hold. In this paper, we systematically verify these assumptions and related optimization analyses across various deep learning tasks using simple, computationally feasible methods. We hope that our findings will serve as a guideline for future research, helping to develop theoretical frameworks that are both analytically tractable and practically applicable.

Importantly, we do not want to just ask "do current assumptions apply to deep neural networks". Instead, we wish to ask whether the *analyses* based on currently prevalent techniques can predict current practical performance. This is a subtly different question: it turns out that most modern analyses actually rely on a few key identities. These identities are usually *empirically measurable* from the iterates of an optimizer. In theoretical analysis, these identities are controlled via various global assumptions (such as convexity or smoothness), but we instead measure directly these identities. This has a significant advantage: not only can it falsify the global assumptions, it can tell if any *different* assumptions can be made that would "rescue" the analysis.

We propose simple, on-the-fly measures that capture how well modern analyses describe practice. We measure these on a wide range of tasks, including basic convex optimization problems, image classification tasks using deep residual networks, and pre-training large language models (LLMs). Overall, our results suggest that most analytical techniques do *not* describe practical performance. Our work fits into a recent trend of challenging and moving past classical optimization assumptions

Simsekli et al. (2019); Zhang et al. (2020b;a); Davis et al. (2021; 2020). However, our focus is not on algorithm development. Instead, we simply want to promote empirical verification of optimization analysis.

Of independent interest, we develop a new smoothness measure closely approximating the sharpness measure. This is an exciting finding, as our measure is computationally feasible even for very deep networks, where computing sharpness is infeasible. This allows for the use of our smoothness measure in studying flat/sharp minima and their implications for generalization in much larger networks. Finally, we offer alternative theoretical analyses for cases where common theoretical assumptions do not hold.

Overall, we feel that our findings motivate two actions in the research community: first, it is important to develop new assumptions and analytical techniques to understand modern optimization. Second, we advocate for verifying any new assumptions by carefully measuring quantities that *actually appear in the optimization analysis* rather than attempting to verify global assumptions.

## 2 BACKGROUND AND EXPERIMENT SETUP

In typical optimization analysis for machine learning, the goal is to minimize objective $F$ given by $F(\mathbf{x}) = \mathbb{E}_{z \sim P_z}[f(\mathbf{x}, z)]$, where $f(\mathbf{x}, z) : \mathbb{R}^d \times \mathcal{Z} \mapsto \mathbb{R}$ is a differentiable function of $\mathbf{x} \in \mathbb{R}^d$. $\mathbf{x}$ indicates the model parameters, $z \in \mathcal{Z}$ indicates an example data point or minibatch of examples, and $P_z$ is some data distribution. The function $F$ represents either a train loss or a population loss depending on various details of the problem setup.

The most common paradigm in optimization analysis is the following three-step strategy: first, identify a "convergence criterion" of interest - for example the loss of some weights output by an algorithm minus the loss of the optimal weights. Second, identify an algebraic expression that can be related to this convergence criterion (often through use of some assumption on the loss landscape). Finally, establish that a given algorithm can guarantee a bound on this algebraic expression (often again using some assumption on the loss landscape):

$$\underbrace{\text{Convergence Criterion}}_{\text{e.g. } \frac{1}{T}\sum_{t=1}^{T} F(\mathbf{x}_t) - F(\mathbf{x}_\star)} \leq \underbrace{\text{Algebraic Expression}}_{\text{e.g. } \frac{1}{T}\sum_{t=1}^{T} \langle \nabla F(\mathbf{x}_t), \mathbf{x}_t - \mathbf{x}_\star \rangle} \leq \underbrace{\text{Upper Bound}}_{\text{e.g. } O(1/\sqrt{T})} \quad (1)$$

The example values for the three terms above are typical of analysis of SGD for convex objectives, in which $\mathbf{x}_\star = \operatorname{argmin} F$, and the middle "algebraic expression" is often termed the *regret* (see Orabona (2019); Hazan (2022) for details).

This paradigm is used in two different ways: first, from a *scientific* perspective, one can try to prove convergence properties for well-known algorithms such as AdamW to explain why these algorithms work well in practice (see e.g. Li & Orabona (2019); Faw et al. (2022); Ward et al. (2020); Zaheer et al. (2018b); Reddi et al. (2019)). Second, from an *engineering* perspective, one can try to design better optimizers from first principles. For this second use-case, the typical approach is to identify a large class algorithms, such as SGD parametrized by the learning rate, and then choose the member of this class that analytically minimizes the upper bound (see e.g. Duchi et al. (2010); McMahan & Streeter (2010); Hazan et al. (2007); Ghadimi & Lan (2013)). This exact approach is how the AdaGrad family of algorithms (which was the intellectual precursor to Adam) was developed.

In order for this paradigm to provide meaningful answers, we must believe that the inequalities in equation (1) hold at least approximately. We can investigate this from two angles: first, we can ask whether the original assumptions that motivated the analysis hold. Second, we can often *empirically measure* expressions related to those appearing in (1), and check the degree to which the desired inequalities hold. These are more likely to hold than the underlying assumptions, because the assumptions imply the inequalities, but the reverse may not be true.

Empirical verification of these inequalities is made especially attractive for two reasons. First, many optimization analyses actually use only a few options for the "algebraic expression" in (1): the only thing that changes is the analysis of the algorithm leading to improved upper bounds. Thus, by empirically measuring the degree to which the *first* inequality in (1) holds, we can interrogate whether popular analyses strategies can explain optimization success in deep learning in way that is less tightly coupled to whether particular global assumptions hold or not.

Two very popular assumptions about the loss landscape and the optimization process are smoothness and convexity. Formally, a differentiable function $f(\cdot, \cdot) : \mathbb{R}^d \times \mathcal{Z} \mapsto \mathbb{R}$ is convex if it satisfies:

$$f(\mathbf{y}, z) \geq f(\mathbf{x}, z) + \langle \nabla f(\mathbf{x}, z), \mathbf{y} - \mathbf{x} \rangle \quad \forall \mathbf{x}, \mathbf{y} \in \mathbb{R}^d, z \in \mathcal{Z}$$

Further, $f(\cdot, \cdot)$ is $L-$smooth if it satisfies:

$$\|\nabla f(\mathbf{x}, z) - \nabla f(\mathbf{y}, z)\| \leq L\|\mathbf{x} - \mathbf{y}\| \quad \forall \mathbf{x}, \mathbf{y} \in \mathbb{R}^d, z \in \mathcal{Z}$$

These are some of the most common assumptions in optimization theory (Zinkevich, 2003; Duchi et al., 2010; Ghadimi & Lan, 2013; Bubeck et al., 2015; Carmon et al., 2017; Zhao et al., 2020; Hu et al., 2019; Hazan, 2022; Cutkosky & Orabona, 2019). We would like to quantify them in our experiments. Since computing the global smoothness constant as well as the convexity of the true loss functions $F(\mathbf{x})$ is infeasible, we instead measure proxies that we call the *instantaneous convexity gap*, denoted by inst_gap, and *instantaneous smoothness*, denoted by inst_smooth, to estimate the levels of convexity and smoothness of the true loss function. Formally, the instantaneous convexity gap with respect to a reference point $\mathbf{y}_t$ of the function $f(\cdot, z_t)$ (stochastic loss function computed at iteration $t$ using datapoint $z_t$) is defined as:

$$\text{inst\_gap}_t(\mathbf{y}_t) := f(\mathbf{x}_t, z_t) - f(\mathbf{y}_t, z_t) - \langle \nabla f(\mathbf{x}_t, z_t), \mathbf{x}_t - \mathbf{y}_t \rangle \tag{2}$$

In our measurements, we use two settings for $\mathbf{y}_t$. First, we consider $\mathbf{y}_t = \mathbf{x}_{t-1}$ to analyze the properties of consecutive points and their impact on the optimization path. Next, we use the constant value $\mathbf{y}_t = \mathbf{x}^\star$, where $\mathbf{x}^\star$ is the *final* iterate produced by a previous training run. This setting provides a more *global* view of the loss landscape. If $f$ is convex, then the convexity gap defined in eq. (2) should be *non-positive*. We also compute the average convexity gap and the exponential moving average of the convexity gaps with respect to a sequence of reference points $\mathbf{y}_1, \dots, \mathbf{y}_t$ (denoted as $\mathbf{y}_{1:t}$ for short), respectively defined as

$$\text{avg\_gap}_t(\mathbf{y}_{1:t}) = \frac{1}{t} \sum_{i=1}^{t} \text{inst\_gap}_i(\mathbf{y}_i),$$
$$\text{exp\_gap}_t(\mathbf{y}_{1:t}) = \beta \cdot \text{exp\_gap}_{t-1}(\mathbf{y}_{1:t-1}) + (1 - \beta) \cdot \text{inst\_gap}_t(\mathbf{y}_t). \tag{3}$$

where $\beta \in (0, 1)$ (we choose $\beta = 0.99$ for our measurements).

Next, we define the instantaneous smoothness at iteration $t$ with respect to $\mathbf{y}_t$ as:

$$\text{inst\_smooth}_t(\mathbf{y}_t) = \frac{\|\nabla f(\mathbf{x}_t, z_t) - \nabla f(\mathbf{y}_t, z_t)\|}{\|\mathbf{x}_t - \mathbf{y}_t\|} \tag{4}$$

If the loss function is $L$-smooth, then $\text{inst\_smooth}_t \leq L$ for all $t \in [T]$. Thus, if this instantaneous smoothness quantity is uniformly bounded by a constant, it could indicate that our loss landscape is smooth. Similar to the convexity gap, we also keep track of other forms of the smoothness measure such as the maximum smoothness and the exponential average smoothness, respectively defined as

$$\text{max\_smooth}_t(\mathbf{y}_{1:t}) = \max_{i \leq t} \text{inst\_smooth}_i(\mathbf{y}_i),$$
$$\text{exp\_smooth}_t(\mathbf{y}_{1:t}) = \beta \cdot \text{exp\_smooth}_{t-1}(\mathbf{y}_{1:t-1}) + (1 - \beta) \cdot \text{inst\_smooth}_t(\mathbf{y}_t). \tag{5}$$

Our maximum smoothness is similar to the smoothness metric proposed in (Santurkar et al., 2018; Zhang et al., 2019). However, instead of tracking the largest smoothness value along the line of the update difference $\mathbf{x}_t - \mathbf{x}_{t-1}$, we keep track of the largest value across all iterations.

Most of our training runs involve multiple epochs. In this case, for the non-instantaneous metrics, we "reset" the averages at the start of each epoch so that the averages contain only iterates from the current epoch. The only exceptions are our pre-training tasks for BERT and GPT-2. Due to the large size of the datasets used in these tasks, we completed the training without traversing the entire dataset. Hence, we do not reset our metrics in these experiments. Beyond smoothness and convexity, we also track many other key properties. We defer these results to the Appendix. These metrics collectively offer deeper insights into the dynamic behavior of the loss function throughout the optimization process.

We conduct experiments across a diverse array of tasks, ranging from simple convex problems to complex NLP tasks involving models with hundreds of millions of parameters. For convex tasks, we run gradient descent on a synthetic dataset using squared loss and also perform logistic regression on various OpenML datasets (Aloi, Connect-4, Covertype, Poker). In the realm of non-convex tasks,

we address both Image Classification and NLP benchmarks. For Image Classification tasks, we train popular benchmark datasets Cifar10 and Imagenet (Deng et al., 2009) on Resnet18 (He et al., 2016) using SGD with momentum (SGDM) and Adamw. we use the configurations reported in (Yao et al., 2020; Tran & Cutkosky, 2022a). For NLP tasks, we pre-train Bert (Devlin et al., 2018b) using the C4 dataset (Raffel et al., 2019) and GPT2 (Radford et al., 2019) using the Pile dataset (Gao et al., 2020). Both tasks are trained using SGDM and AdamW. The learning rates for each optimizer are fine-tuned through a grid search in the range $[10^{-6}, 0.1]$.

## 3  MEASURING CONVEXITY

Convexity is a fundamental assumption in optimization theory since convex functions have many pleasant theoretical guarantees. For instance, every local minimum of a convex function is also a global minimum, which allows us to derive bounds on the suboptimality gap (Bottou & Bousquet, 2007; Defazio et al., 2014; Cutkosky, 2019). Unfortunately, the landscape of deep learning training is known to be non-convex (Jain et al., 2017; Li et al., 2018; Garipov et al., 2018; Choromanska et al., 2015) due to the complex architectures of deep learning models and the nonlinearity of the activation functions. However, the

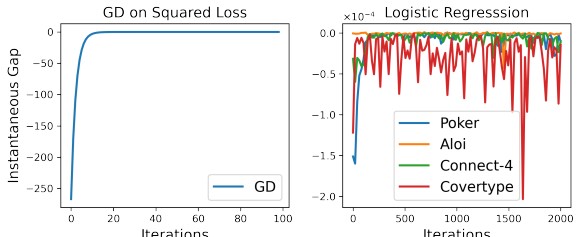

Figure 1: Instantaneous convexity gap w.r.t. $\mathbf{y}_t = \mathbf{x}_{t-1}$ of Gradient Descent (GD) on the squared loss (left) and Logistic Regression on OpenML datasets (Vanschoren et al., 2013) (right).

degree of non-convexity in practical scenarios still remains a bit of a mystery. In this section, we aim to quantify the level of convexity across various machine learning tasks. As a sanity check, we first examine the instantaneous convexity gaps with respect to the previous iterate in simple tasks for which the objective is indeed convex to verify that they align with our theoretical expectations. Results are presented in Fig.1. As we can see from Fig.1, the convexity gap is always non-positive as expected. Now, let us turn our attention to more complex deep learning tasks.

### 3.1  ARE DEEP LEARNING LOSS LANDSCAPES CONVEX ALONG OPTIMIZATION PATHS?

In this section, we aim to examine the convexity along the paths taken by popular optimizers such as Adam and SGD. To achieve this, we compute both the average and the exponential average convexity gaps with respect to the previous iterates, i.e., $\mathbf{y}_t = \mathbf{x}_{t-1}$, across various deep learning benchmarks. Setting $\mathbf{y}_t = \mathbf{x}_{t-1}$, allows us to measure convexity on a "small scale" along the optimization path, rather than as a global property. The presence of any positive gap would indicates non-convexity.

By measuring average gaps (both avg_gap and exp_gap), we gain insight into whether the optimization path could be in some sense "mostly" convex - i.e. whether instantaneous non-convexity is essentially a "rare event". Stochastic optimization analysis typically involves summing or averaging identities derived from convexity, and so one might hope that it is possible to exploit a non-positive average convexity gap.We also provide the instantaneous gap results in Section D in the Appendix.

Surprisingly, the convexity gap along the optimization trajectories of non-convex tasks is not consistently negative or positive, as demonstrated in fig. 2. For instance, while the convexity gap remains uniformly positive (indicating non-convnexity) during the training of ImageNet on ResNet18, the optimization trajectory in the training of Bert frequently shifts between convex and non-convex regions. Notably, in experiments involving CIFAR-10 and GPT-2, the convexity gap consistently exhibits negative values. Similar phenomenon is also observed in (Xing et al., 2018), where they demonstrated that the loss interpolation $F(\alpha\mathbf{x}_t + (1-\alpha)\mathbf{x}_{t+1})$ of deep neural networks trained on CIFAR-10 by SGD is locally convex.

Negative convexity gaps in our experiments do not necessarily indicate convex loss landscapes (since we only check the convexity gap at the points along the optimization trajectory) but rather suggests that effective optimizers like SGD and ADAM can navigate these landscapes by finding paths that are in some sense "locally convex". Further, as illustrated in Figure 2, the dataset plays a significant

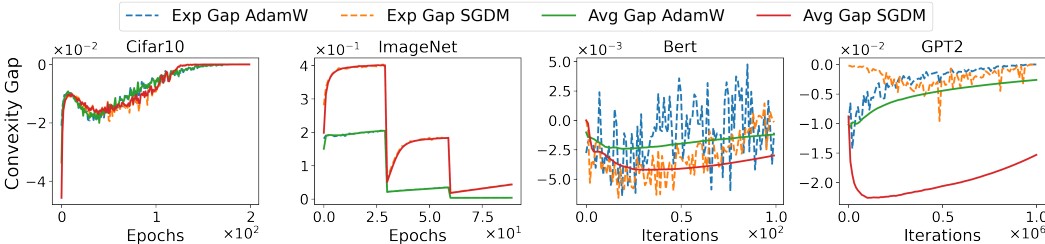

Figure 2: Average convexity gap and exponential average convexity gap w.r.t. $\mathbf{y}_t = \mathbf{x}_{t-1}$ of deep learning benchmarks. In most cases, the gaps are negative, indicating local convexity along training.

role in shaping the loss landscape. Despite using the same optimizer settings and the ResNet-18 architecture, the loss landscapes for the ImageNet and CIFAR-10 datasets show markedly different levels of convexity.

## 3.2 CAN CONVEXITY-BASED ANALYSIS EXPLAIN OPTIMIZATION SUCCESS?

Though the results in Section 3.1 suggest convexity along the optimization path often occurs, we might care more about global convexity, as this is useful to prove global convergence guarantees. Moreover, while the convexity gap can be used to falsify convexity or give intuition about the local properties of the loss landscape, this quantity does not appear in an obvious way in most optimization analyses. So, in this section, we measure a different quantity called *convexity ratio*, which allows us to probe more directly the degree to which analyses based on convexity apply to real problems.

$$\text{convexity\_ratio}_T = \frac{\sum_{t=1}^{T}\langle\nabla F(\mathbf{x}_t), \mathbf{x}_t - \mathbf{x}^\star\rangle}{\sum_{t=1}^{T} F(\mathbf{x}_t) - F(\mathbf{x}^\star)} \tag{6}$$

Here we use a large batch loss to approximate $F$ in cases where it is computationally infeasible to compute $F$ exactly (more details on this computation are in the Appendix). $\mathbf{x}^\star$ is an approximate stationary point given by the output of a previous training run. When $F$ is convex, we should expect the convexity ratio to be larger than 1 so that we have the following important inequality:

$$\sum_{t=1}^{T} F(\mathbf{x}_t) - F(\mathbf{x}^\star) \leq \sum_{t=1}^{T}\langle\nabla F(\mathbf{x}_t), \mathbf{x}_t - \mathbf{x}^\star\rangle \tag{7}$$

Equation (7) is the essential ingredient in many optimization analyses based on convexity. In fact, many analyses of SGD and related methods actually prove convergence by upper-bounding the RHS of the above equation - it is the standard instantiation of eq. (1) for convex analysis (Duchi et al., 2010; McMahan & Streeter, 2010; Zinkevich, 2003; Reddi et al., 2018; Hazan et al., 2007; 2006). For example, a typical analysis of SGD (e.g. (Zinkevich, 2003)) would show that the RHS is bounded by $O(\sqrt{T})$, from which one can then conclude that $\frac{1}{T}\sum_{t=1}^{T} F(\mathbf{x}_t) - F(\mathbf{x}^\star) \leq O(1/\sqrt{T})$: that is, the loss values of the iterates are "on average" approaching the loss of $F(x_\star)$. This holds for all possible values of $x_\star$, even though we will only evaluate it for one particular point.

As a result, even if our function does not satisfy eq. (7), it is still possible to derive global convergence. Assume that the convexity ratio is larger than $K$ for $0 < K < 1$ instead (this condition would be implied by "weak quasi-convexity" studied by Orabona & Tommasi (2017)). Then we still have:

$$\sum_{t=1}^{T} F(\mathbf{x}_t) - F(\mathbf{x}^\star) \leq \sum_{t=1}^{T} \frac{1}{K}\langle\nabla F(\mathbf{x}_t), \mathbf{x}_t - \mathbf{x}^\star\rangle$$

Since our analysis typically bounds the RHS of this equation, the convergence bound degrades by only a factor of $1/K$. Therefore, as long as $K \geq \Omega(1/\sqrt{T})$, popular algorithms like SGD can still ensure global convergence. Interestingly, our experiments on CIFAR-10 and Bert (Figure 3) suggest that this property may hold for certain deep learning tasks. Despite the fact that CIFAR-10 and Bert losses are *not* globally convex, the standard *analysis* used with convexity assumptions may still explain optimization success in these tasks.

For the CIFAR-10 experiments, AdamW's convexity ratio suggests the optimization trajectory remains globally convex relative to the stationary point. While SGDM shows slight non-convexity

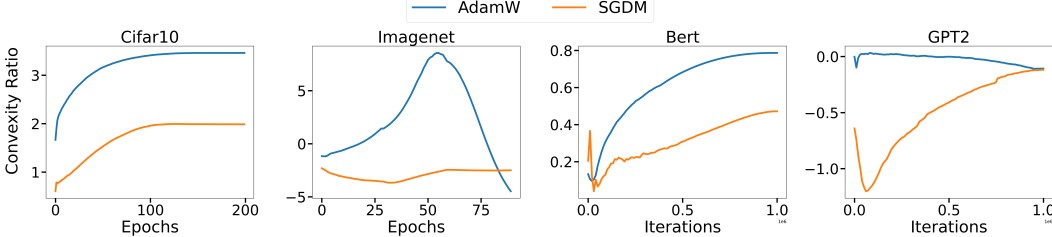

Figure 3: Convexity ratios of deep learning benchmarks. A convexity ratio greater than 1 indicates a convex function. Ratios between 0 and 1 suggest slight non-convexity, still permitting the application of classic convex optimization arguments. Ratios less than 0 denote strong non-convexity.

initially, its convexity ratio consistently exceeds 0.5, allowing for the application of classical convex analysis arguments. In the BERT experiments, both AdamW and SGDM exhibit convexity ratios below 1, indicating a globally non-convex trajectory. However, since the ratios are above 0.1, classical convex analysis remains applicable, though with a 10x degradation in convergence bounds.

Unfortunately, since the convexity ratios of both optimizers are negative in the GPT2 experiments, the convex analysis argument seems to be invalid. A similar lack of convexity is observed in the ImageNet experiments. Interestingly, AdamW seems to often find a "more convex" optimization path compared to SGDM. Nevertheless, these data suggest that significant alterations to classical analysis based on convexity would be needed to adequately explain optimization success for deep learning in general.

## 4  MEASURING SMOOTHNESS

Smoothness assumptions plays a pivotal role in optimization theory. In convex optimization, smoothness can help accelerate the training process and achieve superlinear convergence rate if the loss is strictly convex or strongly convex (Nesterov et al., 2018). In non-convex optimization, smoothness is the key ingredient that makes many convergence analyses possible (Ghadimi & Lan, 2013; Allen-Zhu & Hazan, 2016; Jain et al., 2017; Reddi et al., 2019). Although smoothness is assumed for the majority of non-convex optimization results, it is unclear how well these smoothness conditions are satisfied in practice.

In fact, from a purely theoretical point of view, it may seem unlikely that the objective could be truly smooth: common activation functions such as the ReLU, and common layers such as MaxPools are not globally differentiable and so cannot possibly be smooth. However, one might hope that such issues are essentially pathological problems that do not affect practice. In this section, we attempt to measure smoothness along the real optimization trajectory in an efficient way analogous to our investigation of convexity in Section 3.

We will focus on the exponential average smoothness and the max smoothness defined in eq. (5) since they provide insights into the smoothness level of local and global loss landscape respectively.

First, we compute these measures using the optimally tuned learning rate and schedule in each deep learning experiment. As we can see from fig. 4 (top), in all experiments, the smoothness constants appear to be upper-bounded. However, in many cases these constants are quite large ($10^3$ to $10^6$), making it hard to consider the loss landscapes in these experiments to be smooth in practice. Furthermore, we note that smoothness correlates with changes in the learning rate scheduler. For example, as the learning rate approaches zero at the end of training, the smoothness value increases, as observed in Cifar10 with cosine decay and BERT with linear decay. Similarly, for Imagenet, where we used a piecewise linear scheduler, smoothness increases whenever the learning rate decreases. This observation suggests that smaller learning rates tend to result in larger smoothness values.

To better understand the loss landscapes, we reran all experiments with a constant learning rate (fig. 4 bottom). With constant learning rates, the loss landscape appeared smoother and more stable. Both the max and exponential average smoothness followed a similar pattern: a rapid drop initially (except for SGDM on ImageNet), followed by a consistent rise until reaching a boundary, then stabilizing. Adam typically achieved smaller (i.e., smoother) measures with a learning rate scheduler, while

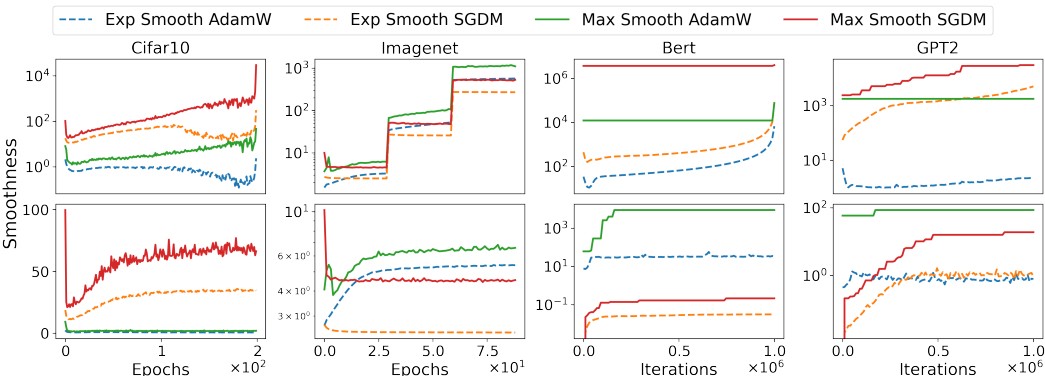

Figure 4: Smoothness measures w.r.t. $\mathbf{y}_t = \mathbf{x}_{t-1}$ of deep learning benchmarks using the optimal configurations. (Top) are the experiments with optimal learning rate scheduler, and (bottom) are the experiments with constant learning rate. Details of experiment setup can be found in Appendix B.

SGD found smaller measures with a constant rate. We conjecture that this phenomenon suggests that SGD's optimization path is more sensitive to changes in the learning rate, while Adam remains robust across different learning rate settings.

### 4.1 SMOOTHNESS MEASURES AS PROXIES FOR SHARPNESS

As shown in fig. 4, the smoothness measured in most experiments exhibit similar behaviors. This pattern closely resembles the edge-of-stability phenomenon observed by (Cohen et al., 2020; 2022) in full-batch SGD and full-batch Adam for smaller tasks. Specifically, Cohen et al. (2020) defines the "sharpness" as the operator norm of the Hessian $\nabla^2 F(\mathbf{x}_t)$. They observe that when training with full-batch gradient descent on CIFAR-10, the sharpness increases until it reaches a value inversely proportional to the learning rate, and then stabilizes.

Our measurements track different quantities than the sharpness, but are faster to compute. Thus, these observations pose an interesting question: *Can our new metrics, max_smooth and exp_smooth, be used as proxies for the sharpness*? If this is true, our approach could substantially expedite the evaluation of sharpness. Our method also makes evaluating the sharpness of much larger models possible (for which computing Hessian information is prohibitively expensive).

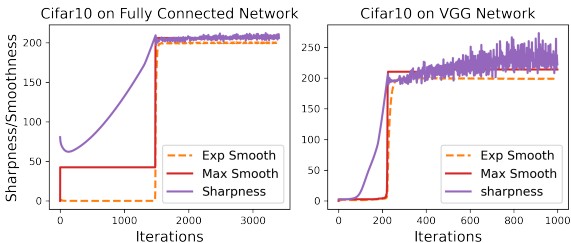

Figure 5: Sharpness (maximum eigenvalue of the training loss Hessian Matrix) v.s. Smoothness.

As discussed above, we notice that a smaller learning rate results in a larger smoothness value. We can potentially explain this using the edge-of-stability phenomenon. (Cohen et al., 2020; 2022) observe that the sharpness is oscillating at the value $c/\eta$ for some constant $c > 0$ and $\eta$ is the learning rate at the edge of stability. Thus, when the learning rate scheduler is applied, any time the learning drops, this boundary increases and causes the smoothness/sharpness level to increase. This phenomenon is also observed in (Cohen et al., 2022). To verify our conjecture, we replicate the experiments in (Cohen et al., 2020) where we train Cifar10 on a simple linear network with tanh activation and on a VGG-11 network (Simonyan & Zisserman, 2014) in Fig.5.

Our new smooth metrics track the actual sharpness value very closely (Fig.5). One possible justification for this is when we measure $\frac{\|\nabla f(\mathbf{x}_t, z) - \nabla f(\mathbf{x}_{t-1}, z)\|}{\|\mathbf{x}_t - \mathbf{x}_{t-1}\|}$, we are effectively estimating how quickly the gradient of the function changes, which is bounded by the Hessian's spectral norm in smooth functions. A higher value indicates a steeper change in the gradient, implying a larger maximum eigenvalue of the Hessian matrix, hence a higher "sharpness". Thus, this metric and the sharpness are inherently related to characterizing the function's smoothness and curvature.

## 4.2 CAN SMOOTHNESS-BASED ANALYSIS EXPLAIN OPTIMIZATION SUCCESS?

The smoothness measurements discussed above are not actually the best criterion for judging the applicability of smooth non-convex optimization analysis. This is because they only capture gradient behavior rather than linking gradients o function values. In typical smoothness-based analysis, one encounters the quantity $\langle \nabla f(\mathbf{x}_{t+1}, z_{t+1}), \mathbf{x}_{t+1} - \mathbf{x}_t \rangle$. In almost all analyses of non-convex optimization algorithms, this quantity usually plays the role of the "algebraic expression" in (1) (Khaled & Richtárik, 2020; Li et al., 2024; Zaheer et al., 2018a; Carmon et al., 2018; Li & Orabona, 2019; Faw et al.,

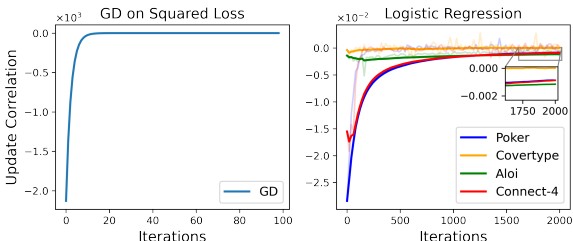

Figure 6: Update correlation of GD on the squared loss (left) and logistic regression on OpenML datasets (right). The blurred lines are the actual update correlations, and the thick lines are the varage.

2022; Reddi et al., 2019). To illustrate, consider an optimizer with update $\mathbf{x}_{t+1} = \mathbf{x}_t + \Delta_t$, and assume that $F$ is $L$-smooth, $\mathbb{E}[\Delta_t] = -\eta \nabla F(\mathbf{x}_t)$ and $\mathbb{E}[\|\Delta_t\|^2] \leq \eta^2 G^2$. Then:

$$\mathbb{E}[F(\mathbf{x}_{t+1}) - F(\mathbf{x}_t)] \leq \mathbb{E}[\langle \nabla F(\mathbf{x}_t), \mathbf{x}_{t+1} - \mathbf{x}_t \rangle + \tfrac{L}{2}\|\mathbf{x}_{t+1} - \mathbf{x}_t\|^2]$$

$$\leq -\eta \, \mathbb{E}\left[\|\nabla F(\mathbf{x}_t)\|^2\right] + \tfrac{L\eta^2 G^2}{2}. \tag{8}$$

Typical analyses show that $-\eta \, \mathbb{E}[\|\nabla F(\mathbf{x}_t)\|^2]$ dominates $\frac{L\eta^2 G^2}{2}$ so that $F(\mathbf{x}_t)$ decreases over time. Intuitively, this holds if we make $\eta$ sufficiently small because the negative term is linear in $\eta$ while the positive term is quadratic in $\eta$. Note that this high-level idea is used even for analyses based on less classical smoothness assumptions such as (L0,L1) smoothness (Zhang et al., 2019).

To check whether this analysis technique can explain the success of practical optimizers, we would like to measure the inner-product $\langle \nabla F(\mathbf{x}_t), \mathbf{x}_{t+1} - \mathbf{x}_t \rangle$ and see if it is negative. This would directly capture the optimization analysis because in the typical analysis, all of the provable decrease in the function value is caused negative inner-products.

Unfortunately, this inner-product is difficult to estimate empirically because we do not know $\nabla F(\mathbf{x}_t)$. One might consider instead estimating it using $\langle \nabla f(\mathbf{x}_t, z_t), \mathbf{x}_{t+1} - \mathbf{x}_t \rangle$. However, this approach is flawed because $\mathbf{x}_{t+1} - \mathbf{x}_t$ is not independent of $z_t$, giving the correlation a negative bias. Instead, we measure a quantity that we call the *update correlation*, which is defined as

$$\text{update\_corr}_t := \langle \nabla f(\mathbf{x}_{t+1}, z_{t+1}), \mathbf{x}_{t+1} - \mathbf{x}_t \rangle. \tag{9}$$

Since $\mathbf{x}_{t+1} - \mathbf{x}_t$ is independent of $z_{t+1}$, the update correlation is an unbiased estimator of $\langle \nabla F(\mathbf{x}_{t+1}), \mathbf{x}_{t+1} - \mathbf{x}_t \rangle$. Moreover, it turns out that update correlation still captures the same notion of "function" progress measured by typical analysis. Here's a brief reasoning. Consider the update $\mathbf{x}_{t+1} = \mathbf{x}_t + \Delta_t$ and assume $F$ is $L$-smooth (but this time we don't make assumptions on $\Delta_t$). By smoothness,

$$F(\mathbf{x}_{t+1}) - F(\mathbf{x}_t) \geq \langle \nabla F(\mathbf{x}_{t+1}), \mathbf{x}_{t+1} - \mathbf{x}_t \rangle - \tfrac{L}{2}\|\mathbf{x}_{t+1} - \mathbf{x}_t\|^2 \tag{10}$$

$$F(\mathbf{x}_{t+1}) - F(\mathbf{x}_t) \leq \langle \nabla F(\mathbf{x}_{t+1}), \mathbf{x}_{t+1} - \mathbf{x}_t \rangle + \tfrac{L}{2}\|\mathbf{x}_{t+1} - \mathbf{x}_t\|^2 \tag{11}$$

Consequently, if $\langle \nabla F(\mathbf{x}_{t+1}), \mathbf{x}_{t+1} - \mathbf{x}_t \rangle$ is negative, for small enough learning rates $\eta$ the global loss decreases and the optimizer is consistently making progress. On the other hand, a positive update correlation $\langle \nabla F(\mathbf{x}_{t+1}), \mathbf{x}_{t+1} - \mathbf{x}_t \rangle$ appears to be disastrous since this analysis would suggest that the loss should increase. In particular, we are not aware of any analysis based upon negative values of $\langle \nabla F(\mathbf{x}_t), \mathbf{x}_{t+1} - \mathbf{x}_t \rangle$ that does not also predict negative values for the update correlation. Therefore, if the standard analysis of smooth non-convex optimization can explain optimization success in deep learning, then in every experiment we should expect that the update correlation $\langle \nabla f(\mathbf{x}_{t+1}, z_{t+1}), \mathbf{x}_{t+1} - \mathbf{x}_t \rangle$ should be negative on average.

First, we check if this is the case for simple convex experiments (fig. 6). In all of these experiments, the update correlations are negative on average, which agrees with our intuition that a negative update correlation indicates progress in the training.

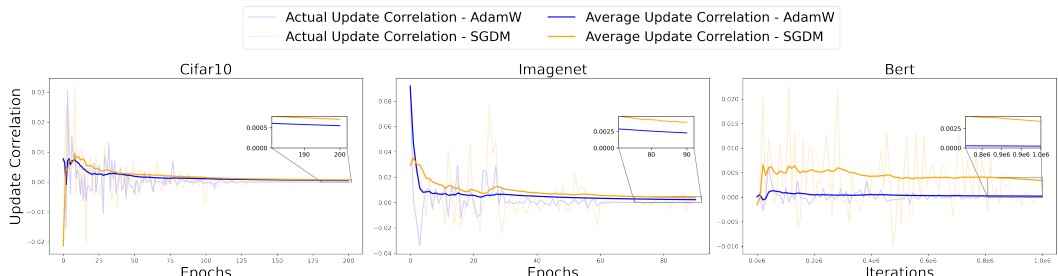

Figure 7: Update correlation on various Deep Learning tasks.

However, surprisingly, the update correlation is positive on average in almost every other Deep Learning experiment (fig. 7). This is a fascinating phenomenon because it indicates that the optimizer changes direction very often, and yet it still effectively minimizes the loss. This suggests that the classic smooth non-convex analysis that relies on the descent lemma is problematic in practice. The only case of negative correlations is GPT-2 on the Pile dataset, but they turn positive when the dataset is shuffled or replaced with the C4 dataset. It would be interesting to find out exactly the cause of this behavior.

The observation that $\nabla F(\mathbf{x}_{t+1})$ is positively correlated with $\mathbf{x}_{t+1} - \mathbf{x}_t$ suggests that the objective may be "poorly conditioned", so that the optimizer is bouncing back-and-forth along the walls of a narrow ravine in the optimization landscape. Previous empirical studies have also suggested similar dynamics (Rosenfeld & Risteski, 2023). The classical mitigations for poorly conditioned objectives in the *deterministic or convex* settings are preconditioning, including via second-order algorithms, as well as accelerated gradient descent (e.g. Gupta et al. (2018); Liu et al. (2023a); Yao et al. (2021); Nesterov et al. (2018); Dozat (2016)). However, the advantages of such techniques are poorly understood in the stochastic setting (indeed, there is no advantage in the worst-case (Arjevani et al., 2020)). Instead, most current analyses we are aware of in the stochastic setting appear to rely on negative update correlations.

### 4.3 ALTERNATIVES FOR SMOOTH NON-CONVEX OPTIMIZATION

In previous sections, we observed that some common assumptions or identities used in analysis, such as convexity, smoothness, or negative update correlation, might not hold in practice. In this section, we will discuss alternative frameworks that do not rely on these assumptions.

The first direction focuses on a family of weakly convex objectives (Davis & Drusvyatskiy, 2019; Mai & Johansson, 2020), where the goal is to minimize a proxy of the objectives called the Moreau envelope (Moreau, 1965) . In a different direction, Zhang et al. (2020b) propose employing the Goldstein stationary point (Goldstein, 1977) as a convergence criterion that is tractable for non-smooth objectives. Later, Cutkosky et al. (2023) proposes an online-to-non-convex conversion (O2NC) technique that later inspires other works on non-smooth non-convex optimization (Ahn et al., 2024; Zhang & Cutkosky, 2024). The key idea of their technique is the use of random scaling: suppose $s_t$ is sampled i.i.d. from $\mathrm{Exp}(1)$, then $\mathbf{x}_{t+1} = \mathbf{x}_t + s_t \Delta_t$ satisfies

$$\mathbb{E}_{s_t}[F(\mathbf{x}_{t+1}) - F(\mathbf{x}_t)] = \mathbb{E}_{s_t}\langle \nabla F(\mathbf{x}_{t+1}), \Delta_t \rangle. \tag{12}$$

We refer to the update form $\mathbf{x}_{t+1} = \mathbf{x}_t + s_t \Delta_t$ where $s_t \sim \mathrm{Exp}(1)$ i.i.d. as the *update with random scaling (RS)*, and the update with $s_t \equiv 1$ as the *update without RS*. Unlike the lower bound in equation 10, the equality in equation 12 suggests that $\langle \nabla F(\mathbf{x}_{t+1}), \Delta_t \rangle$, which we referred to as *update correlation with RS*, is an unbiased estimator of function progress $F(\mathbf{x}_{t+1}) - F(\mathbf{x}_t)$ and a good indicator of the training progress: we should expect $F(\mathbf{x}_t)$ to decrease as long as $\langle \nabla F(\mathbf{x}_{t+1}), \Delta_t \rangle$ is negative in average.

To verify if the theory holds in practice, we test SGDM and AdamW with random scaling updates and compare them to their counterparts without RS. Specifically, we measure the following three properties: update correlation, update correlation with random scaling, and instantaneous loss difference, where the first is defined in eq. (9) and the latter two are respectively defined as

$$\text{update\_corr\_RS}_t = \langle \nabla f(\mathbf{x}_t, z_t), \Delta_{t-1} \rangle, \ \ \text{loss\_diff}_t = f(\mathbf{x}_t, z_t) - f_t(\mathbf{x}_{t-1}, z_t). \tag{13}$$

Note that if the update does *not* have random scaling applied, then update\_corr\_RS$_t$ = update\_corr$_t$.

In fig. 8 we plot the cumulative sum of these quantities. The sum of update correlation always increases, regardless of whether random scaling is employed. However, for optimizers with random scaling, the sum of update_corr_RS$_t$ decreases and closely aligns with the sum of loss difference. This supports the theory that update_corr_RS$_t$ is an unbiased estimator of loss difference, even for complicated LLM models. Also, it motivates a guideline for developing empirically effective optimizers: keeping $\langle \nabla f(\mathbf{x}_t, z_t), \Delta_{t-1} \rangle$ as negative as possible while applying random scaling to the update.

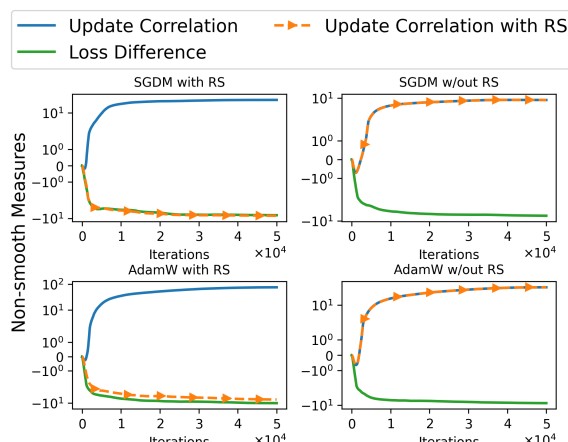

Figure 8: Cumulative sum (symmetric log scale) of update correlation, update correlation with RS, and loss difference of GPT2 model trained on Pile dataset. (Top) is SGDM and (bottom) is AdamW; (left) is update with RS and (right) is the benchmark without RS.

## 5 RELATED WORKS

There have been extensive studies on the empirical properties and the loss landscape of modern machine learning. Goodfellow & Vinyals (2015) proposed one-dimensional and two-dimensional visualization tools for the loss landscape of various neural networks, demonstrating that SGD rarely encounters local minima during training. Im et al. (2017) tested the training trajectories of different optimizers using the same visualization tools and observed that different optimizers exhibit distinct behaviors when encountering saddle points. Li et al. (2018) proposed more refined visualization techniques and showed that the smoothness of the loss landscape closely correlates with generalization performance. Nakkiran et al. (2019) studied the dynamics of SGD training, showing that SGD learns simple classifiers at early training stages and learns more complex classifiers at later stages. Power et al. (2022) reported the grokking phenomenon on a synthesized dataset such that after a long period of severe overfitting, validation score suddenly increases to almost perfect generalization. Thilak et al. (2022) revealed the slingshot effect of training neural networks with adaptive optimizers, which is a cyclic behavior between stable and unstable regimes during training process. While these results provide general insight into neural network landscapes, we focus on validating common assumptions and key identities fundamental to the analysis of optimization theory.

There are several studies that align more closely with our work. Xing et al. (2018) demonstrated that loss interpolation between consecutive iterates is locally convex, which agrees with our observations in Sec 3.1. While their experiments focus on SGD and image classification tasks, we extended the scope of our convexity measures to include AdamW and LLMs. Furthermore, we also tested a more global convexity measure in Sec 3.2. Cohen et al. (2020; 2022) observed the "edge of stability" phenomenon where the sharpness increases during early stage of training and then stabilizes. Our observations in Sec 4 align with their finding and extend beyond CIFAR-10 tasks. Rosenfeld & Risteski (2023) demonstrated the opposing signal phenomenon that there are groups of outliers such that decreasing loss over one group increases loss over other groups, which could explain our observation of positive update correlation in Sec 4.2. Unlike these works, our work does not only verify common assumptions but also directly measures key quantities in modern analyses.

## 6 CONCLUSIONS

We address the critical question of whether modern analyses in stochastic optimization theory align with practice. To this end, we empirically measure key quantities that are commonly used in theory across a diverse range of machine learning benchmarks. Our results indicate that, in most cases, these commonly assumed identities do not hold in practice. Further, we provide comparisons between the behaviors of SGD and Adam across various important properties. We hope that our experiments results can contribute to a better understanding of what enables practical optimization, as well as motivate more rigorous empirical verification of optimization analyses in the future.

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
