## A    LICENSE

Image Classification: Imagenet is distributed under the BSD 3-Clause License, Resnet is distributed under the Apache License, Cifar10 is distributed under the MIT License.

NLP: Bert/GPT2 (Hugging Face), C4 dataset are distributed under the Apache 2.0 License. The Pile dataset is distributed under the MIT License.

## B    EXPERIMENTS SETTINGS AND CONFIGURATIONS

**GD on squared loss:** We run Gradient Descent on squared loss using synthetic datasets. We train the optimizer for 100 iterations with a learning rate equals to 0.1. We report the metrics computed every iteration.

**Logistic regression with OpenML datasets:** We run logistic regression with commonly used OpenML datasets such as Aloi (42396), Poker (1595), Connect-4 (1591), and Covertype (150). We run all experiments using a batch size equal to 64 and AdamW as the optimizer. We tune the learning rates using a grid search over the range $[1e - 4, 1]$. We report the metrics computed at the end of every epoch.

**Training Cifar10 on Resnet18:** We train the benchmark dataset Cifar10 on Resnet18 using SGDM and Adamw. For SGDM, we use a learning rate $= 0.1$ and for AdamW, we use a learning rate $= 0.001$. Both optimizers are trained with batch size equal to 128, weight decay equal to $5e - 4$, and cosine learning rate scheduler. In the experiments with constant learning rates, we use the same optimal configurations as the normal experiments but without the scheduler. We train both optimizers for 200 epochs and all tracking measures (convexity gap, max smoothness, etc,...) are reset for the new epoch (this is why we see the max smoothness quantity goes down at various points in Fig.4). We use full batch to compute the large batch loss ($F(x)$) and gradient $\nabla F(x)$. We report the metrics computed at the end of every epoch.

**Training Imagenet on Resnet18:** We train Imagenet on Resnet18 using SGDM with a learning rate equal to 0.1 and Imagenet with a learning rate equal to 0.001. The weight decay is $1e - 4$ and we employ a learning rate scheduler that decays the learning rate by 10 every 30 epochs for both optimizers. These are the experiments configurations used in (Yao et al., 2020; Tran & Cutkosky, 2022a). Similar to the Cifar10 experiments, we keep the same configurations except for the learning rate scheduler for the constant learning rates experiments. We also reset the tracking quantities every epoch. We use full batch to compute the large batch loss ($F(x)$) and gradient $\nabla F(x)$. We report the metrics computed at the end of every epoch.

**Pre-train Bert using the C4 dataset:** We train the "bert-base-cased" model of HuggingFace (Devlin et al., 2018a) from scratch using the C4 dataset. The model has approximately 110 million trainable parameters. We train the model for 1 million iterations with 10k warm-up steps and a linear decay scheduler. AdamW is trained with a learning rate of $5e - 5$ and SGDM is trained with a learning rate of $1e - 3$. The weight decay is set to be $0.01$ for both optimizers. Since the training never gets through the whole C4 dataset, we do not reset the value of the tracking quantities. For experiments with constant learning rates, we keep the same configurations but without the scheduler and the warm-up step. We use a batch size of 100000 to compute the large batch loss ($F(x)$) and gradient $\nabla F(x)$. We report the metrics computed every 10k iterations.

**Pre-train GPT2 using the Pile dataset:** We train the GPT2 model of HuggingFace (Devlin et al., 2018a) from scratch using the Pile dataset. The model has approximately 124 million trainable parameters. We train the model for 1 million iterations with 10k warm-up steps and a linear decay scheduler. Both SGDM and AdamW are trained with a learning rat of $1e - 4$. The weight decay is set to be $0.01$ for both optimizers. We do not reset the value of the tracking quantities. For experiments with constant learning rates, we keep the same configurations but without the scheduler and the warm-up step. We use a batch size of 100000 to compute the large batch loss ($F(x)$) and gradient $\nabla F(x)$. We report the metrics computed every 10k iterations.

**Testing non-smooth measures:** We train three different tasks with SGDM and AdamW with and without random scaling. We use a variant implementation of SGDM, which updates

$$\Delta_t = \beta(\Delta_{t-1} - \eta_t g_t), \quad x_{t+1} = x_t + s_t \Delta_t.$$

$s_t$ is sampled i.i.d. from $\text{Exp}(1)$ with random scaling turned on, and $s_t \equiv 1$ otherwise. This is equivalent to SGDM with different effective learning rate and momentum constants, and is shown to have theoretical guarantee (Zhang & Cutkosky, 2024). We use the standard implementation of AdamW, with the only difference being the inclusion of the additional random scalar.

In the first task, we train the ResNet18 model on the Cifar10 dataset for 200 epochs with batch size $= 128$, with a total of roughly 80k iterations. For SGDM, we use a learning rate $= 0.01$ and momentum $\beta = 0.99$. For AdamW, we use a learning rate $= 3e - 4$, weight decay $= 0.1$ and default values $b_1 = 0.9, b_2 = 0.999$. For both optimizers, we use linear decay scheduler with 5k warmup steps.

In the second task, we train the "bert-base-cased" model from scratch on the C4 dataset for 50k iterations with 5k warmup steps and a linear decay scheduler. For SGDM, we use a learning rate $= 1e - 3$ and momentum $\beta = 0.99$. For AdamW, we use a learning rate $= 5e - 5$, weight decay $= 0.01$ and default values $b_1 = 0.9, b_2 = 0.999$.

In the third task, we train the GPT2 model from scratch on the Pile dataset for 50k iterations with 5k warmup steps and a linear decay scheduler. For SGDM, weuse a learning rate $= 0.01$ and momentum $\beta = 0.99$. For AdamW, we use a learning rate $= 3e - 4$, weight decay $= 0.1$ and default values $b_1 = 0.9, b_2 = 0.999$. In all tasks, the optimizers with random scaling have the same configuration as its benchmark without random scaling.

**Runtime:** All experiments are run on 1 NVIDIA v100 GPUs. Cifar10 experiments take 3 hours, Imagenet experiments take 58 hours, both GPT2 and Bert experiments take about a week to train.

**Code:** All experiments can be found in the anonymous repository: `https://github.com/Neurips24-Submission14212/Submission14212`.

## C   NOTATIONS AND DEFINITIONS

Below we list all the notations and definitions related to our measurements.

| Symbol | Description |
|---|---|
| $\text{inst\_gap}_t(\mathbf{y})$ | Instantaneous convexity gap in iteration $t$ w.r.t. $\mathbf{y}$, defined in equation 2 |
| $\text{avg\_gap}_t(\mathbf{y}_{1:t})$ | Unweighted average of $\text{inst\_gap}_i(\mathbf{y}_i)$, defined in equation 3 |
| $\text{exp\_gap}_t(\mathbf{y}_{1:t})$ | Exponential average of $\text{inst\_gap}_i(\mathbf{y}_i)$, defined in equation 3 |
| $\text{convexity\_ratio}_t$ | Convexity ratio, defined in equation 6 |
| $\text{inst\_smooth}_t(\mathbf{y})$ | Instantaneous smoothness in iteration $t$ w.r.t. $\mathbf{y}$, defined in equation 4 |
| $\text{exp\_smooth}_t(\mathbf{y}_{1:t})$ | Exponential average of $\text{inst\_smooth}_i(\mathbf{y}_i)$, defined in equation 5 |
| $\text{max\_smooth}_t(\mathbf{y}_{1:t})$ | Maximum over $\text{inst\_smooth}_i(\mathbf{y}_i)$, defined in equation 5 |
| $\text{update\_corr}_t$ | Update correlation in iteration $t$, defined in equation 9 |
| $\text{update\_corr\_RS}_t$ | Update correlation with random scaling in iteration $t$, defined in (13) |
| $\text{loss\_diff}_t$ | Instantaneous loss difference in iteration $t$, defined in (13) |

Table 1: Notations of the key identities measured in our experiments.

## D   EXTRA EXPERIMENTS RESULTS

In this section, we report some results that we do not have space to include in the main text.

### D.1 THE NORM OF THE GRADIENT INCREASES AS THE TRAINING PROGRESSES

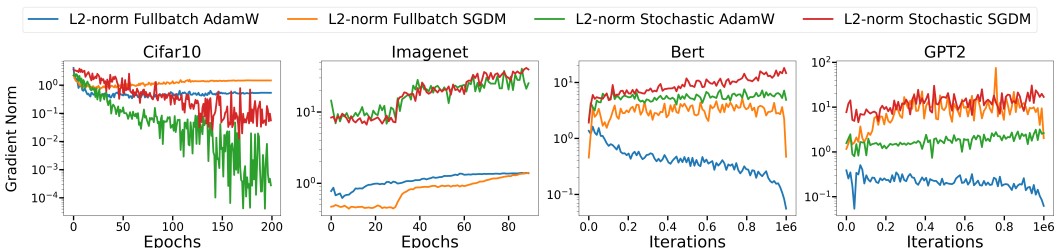

Figure 9: The $L2-$ norm of the gradients as the training progresses.

When the objective is non-convex, since finding the global minima is NP-hard, previous works focus on finding the $\epsilon-$stationary point (Tran & Cutkosky, 2022b; Fang et al., 2018; Arjevani et al., 2020), which is defined as a point such that the gradient $\|\nabla F(\cdot)\| \leq \epsilon$. The common assumption is that an optimizer performs well if it can find points with a small gradient norm, which is expected to decrease as training progresses. However, as we can see from Fig.9, this is not always the case in practice. In Cifar10 and Bert experiments,the full-batch gradient norms decrease for "good" optimizers (SGDM and AdamW for CIFAR-10, and AdamW for BERT), which supports the theory. Conversely, in the Imagenet and GPT2 experiments, the gradient norms hardly decrease, even though the optimizers are still making consistent progress. In fact, in the Imagenet experiments, the norms actually increase, indicating that we are straying further from the stationary point. This suggests that the use of $\epsilon-$stationary point as the convergence criterion might not be appropriate in practice.

### D.2 GRADIENT STANDARD DEVIATION INCREASES

Let us compute the gradient standard deviation as $\sigma := \frac{1}{T} \sum_{t=1}^{T} \|\nabla f(x_t, z_t) - \nabla F(x_t)\|$. Intuitively, the optimizer might make rapid progress if the variance (or standard deviation) is small since it means that our gradient estimate $\nabla f(x_t, z_t)$ is approximating the true gradient well. This is the intuition that leads to the development of a branch of optimization algorithms called variance-reduced algorithms (Allen-Zhu & Hazan, 2016; Cutkosky & Orabona, 2019; Johnson & Zhang, 2013), Thus, we would expect that as the optimizer making progresses, the standard deviation also decreases.

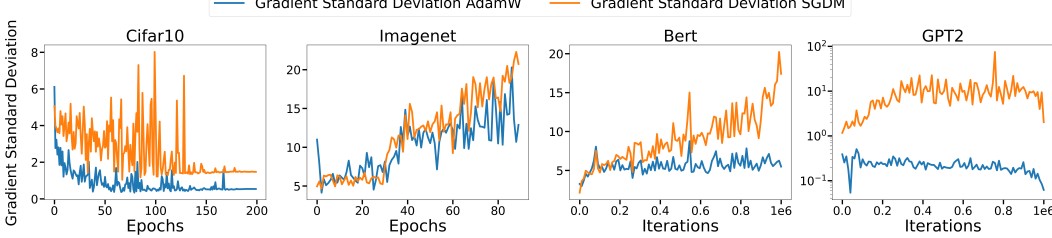

Figure 10: Standard deviation of the gradients

However, similar to the gradient norms, the standard deviation also does not decrease in every experiment. It is hard to conclusively justify why this is the case. One possible explanation for this phenomenon is the existence of multiple minima or low-loss "valley". Thus, even though the optimizer is deviating from the direction to a low-loss "valley" indicating by the true gradient, it is somehow still able to navigate to a different low-loss valley, thus it continues making progress. Further, we note that Adam also consistently returns gradient that is closer to the true gradient. It would be interesting to investigate further to see if this is a property of Adam or of any adaptive method.

## D.3 Parameters Norm

We compute the total parameters norm of the model in each experiment. Adam consistently has larger parameters norm than SGD.

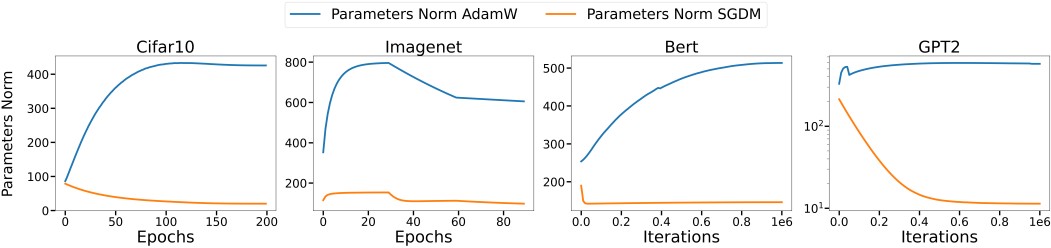

Figure 11: The total $L2-$norm of Model parameters

## D.4 $L1-$norm of the stochastic gradients

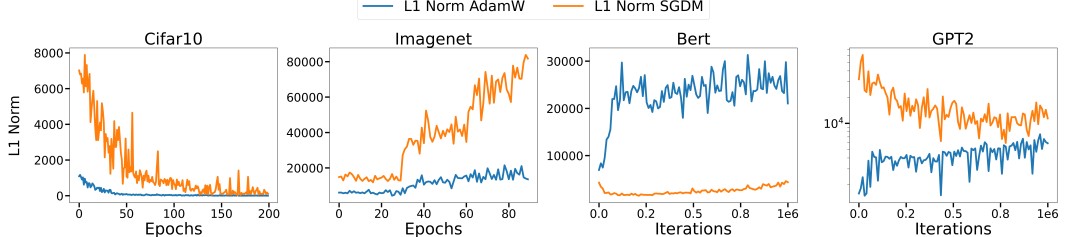

Figure 12: $L1-$norm of the stochastic gradients

We present additional results in the $L1$-norm of the gradient to complement our $L2$-norm findings discussed in Section D.1. An interesting observation is that, although the $L2$-norm of SGD is consistently larger than that of Adam, this is not the case for the $L1$-norm (BERT experiments). This discrepancy suggests that the larger $L2$-norm in SGD may be attributed to outliers in the gradient coordinates, which significantly inflate the final norm. In contrast, Adam, with its adaptive learning rate for each coordinate, effectively minimizes all directions simultaneously, avoiding the issue of gradient outliers.

## D.5 Test accuracy for Image Classification

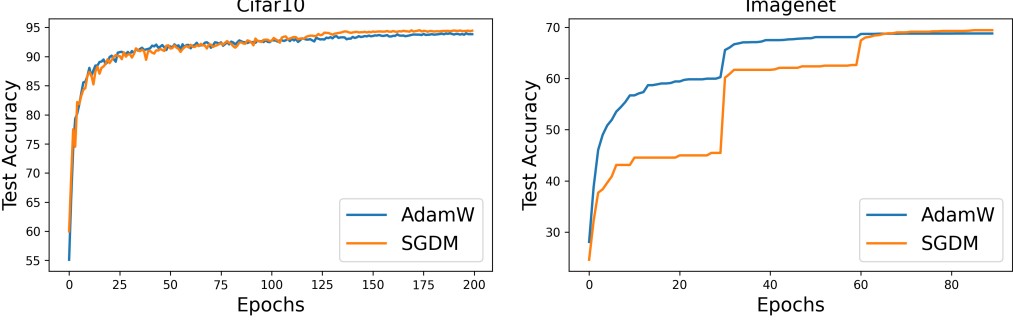

Figure 13: Test Accuracy of Cifar10 and Imagenet trained on ResNet18

### D.6 VALIDATION LOSS OF NLP TASKS

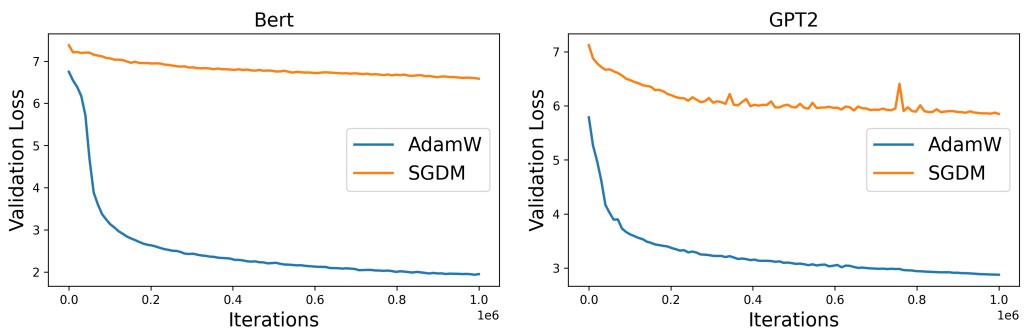

Figure 14: Validation loss of pre-training Bert on C4 and GPT2 on the Pile

### D.7 INSTANTANEOUS CONVEXITY GAP FOR DEEP LEARNING TASKS

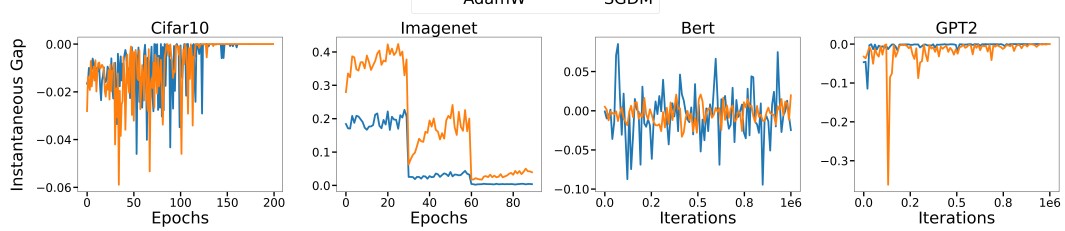

Figure 15: Instantaneous convexity gap w.r.t. $\mathbf{y}_t = \mathbf{x}_{t-1}$ of deep learning benchmarks. Non-positive gap indicates convexity. See Section 3.1 for detailed discussions.

### D.8 UPDATE CORRELATION: SHUFFLED VS UNSHUFFLED

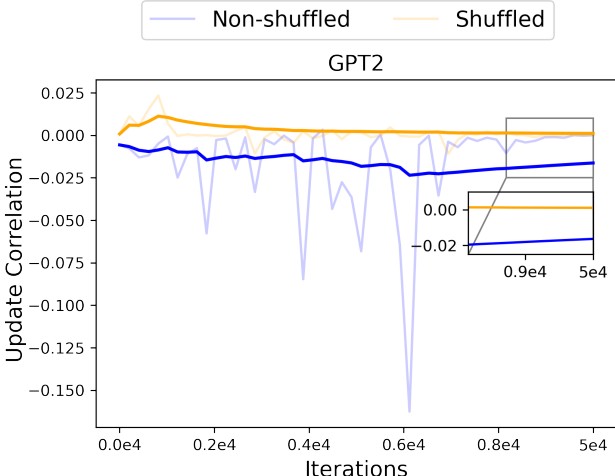

Figure 16: Update correlations of pre-training GPT2 on the Pile dataset - one experiment uses shuffled dataset, the other just iterates through the original dataset. Both are trained for 50k iterations.

## D.9    NON-SMOOTH MEASURES FOR OTHER DEEP LEARNING TASKS

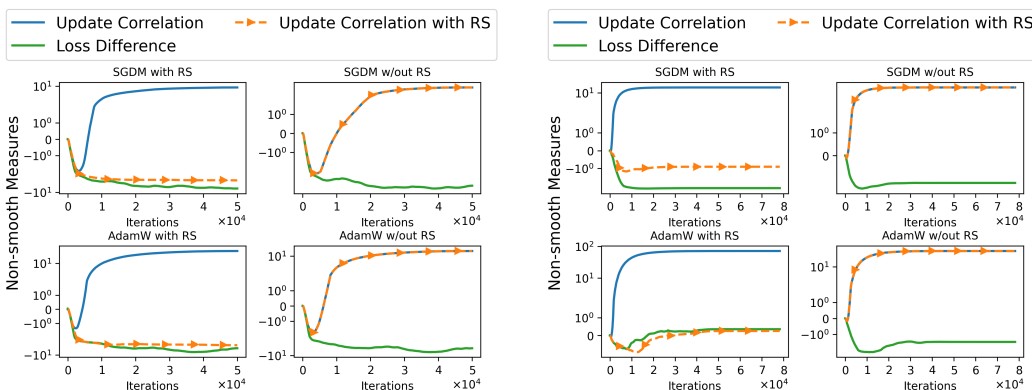

Figure 17: Cumulative sum (symmetric log scale) of update correlation, update correlation with RS, and loss difference of Bert model trained on C4 dataset (left) and ResNet18 model trained on CIFAR10 dataset (right). Top row is SGDM and bottom row is AdamW; left column is update with RS and right column is the benchmark without RS. See Section 4.3 for detailed discussions.