# OpenReview forum: "Reevaluating Theoretical Analysis Methods for Optimization in Deep Learning"
_ICLR.cc/2025/Conference — ICLR 2025 Conference Withdrawn Submission_

### Official Review · Reviewer_rzFo · 2024-11-01

**Soundness:** 3
**Presentation:** 3
**Contribution:** 2
**Rating:** 6
**Confidence:** 3

**Summary:**

This paper empirically tests whether theoretical optimization analyses apply to deep learning optimization problems.  While the global assumptions made in classical analyses such as convexity and smoothness obviously do not literally apply in deep learning, one might hope that classical analyses still can be partly rescued.  The goal of this paper is to test whether that is the case.

In section 3.1, the authors show that the deep learning loss functions often exhibit negative curvature when interpolating between successive iterates, which clashes with convexity.  In section 3.2, they show that a popular convexity-based bound analysis technique could still conceivably apply to some of the tested deep learning optimization problems, but not to others.  In section 4, the authors show that deep learning loss functions are not smooth, and that the smoothness along the trajectory also depends on the learning rate (which is closely related to observations from prior works).  In section 4.1, they point out that their smoothness measure can be viewed as a computationally cheap proxy for Hessian spectral norm.    In section 4.2, they observe that successive updates are negatively correlated, which invalidates a certain class of optimization analyses.  In section 4.3, they explore an alternative analysis techinque.

Overall, the authors conclude that classical optimization analyses are generally inapplicable - "real optimizers often make progress even when typical optimization analysis suggests that they should not."

**Strengths:**

- the findings in section 3.2 imply that a popular convexity-based analysis technique cannot apply in deep learning (since the so-called 'convexity ratio' is negative in many deep learning settings)
- in section 4, it is shown that a certain measure of smoothness (the relative change in the gradients after each step) is large for deep learning landscapes.  This implies that smoothness-based analyses based on this property do not apply in deep learning.  This is a more computationally tractable measure of smoothness than the Hessian spectral norm, which has been considered in previous works, but is computationally expensive for large networks.
- while prior works have reported highly similar findings to those of section 4, experiments in those works did not cover 'modern' architectures such as  GPT2.

**Weaknesses:**

- the quantity measured in section 3.1 (which measures whether the loss is convex along the line between successive iterates) does not really enter into convergence analyses.  Thus, it's not clear what is the significance of these results for convergence analyses.
- the paper's findings regarding smoothness (e.g. that it adapts to the learning rate) are similar to results which have already been reported in prior works, in particular Cohen et al '21.  Prior works such as Xing et al '18 have already reported that successive updates are anti-correlated, which is the main finding of section 4.3.1.  The conclusion that "the classic smooth non-convex analysis that relies on the descent lemma is problematic in practice" (line 446) was already stated in Cohen et al '21, in the discussion section.
- since the loss difference plots in figure 8 are the same with or without random scaling, it's not clear to me whether the guideline mentioned in section 497 should actually be viewed as a guideline for optimizer design.

**Questions:**

- lines 205-212 report that on several networks (e.g. ResNet on ImageNet), the 'instantaneous convexity gap' between successive iterates is often positive during training.  If I understand correctly, this means that the loss at step $t+1$ lies _below_ the local linear approximation of the loss at step $t$.   I find this surprising, and in fact somewhat hard to believe -- I would think that the curvature along the step direction is always positive.  In order to be sure that there is no bug, could the authors please provide a one-dimensional plot of the loss function along the step direction, i.e. a plot of $F(\alpha x_t+(1-\alpha) x_{t+1})$ for $0 \le \alpha \le 1$.
  - If the reported result is correct, I suspect that the difference between CIFAR-10 and ImageNet experiments is due to some factor we could understand such as differences in the batch size, rather than to an intrinsic difference between the CIFAR-10 and ImageNet datasets.
- why didn't you ever assess convexity by tracking the instantaneous convexity gap between the iterate $x_t$ and the ultimate solution $x^*$?
- in figure 5, over what time ranges are you computing the convexity ratios?  is the convexity ratio at epoch $t$ a sum over all $t$'s within the epoch, or a sum over all $t$'s up to and including the epoch?

---

> ### Author Response · Authors · 2024-11-19
>
> Thank you for suggesting the experiment! By doing so, we notice some issues with our imagenet experiments. Therefore, we decided to withdraw our paper to correct the result. Thank you again for your help!

---

### Official Review · Reviewer_b2dZ · 2024-11-03

**Soundness:** 3
**Presentation:** 3
**Contribution:** 2
**Rating:** 6
**Confidence:** 3

**Summary:**

In this paper, the authors numerically track some quantities (related to convexity and smoothness) on the iterates of SGDm and Adam across the training epochs. The values of these quantities (e.g., ≥ 0 or < 0, ≥ 1 or < 1) are important for classical convergence analysis to proceed. Their experimental results might shed light on new theoretical frameworks explaining the success of popular stochastic algorithms used in deep learning.

**Strengths:**

- Overall, the main takeaways of this paper seem to be interesting and systematically validated. While some of them may not be entirely new to the readers, the paper effectively connects the dots to provide a clearer overall picture.
- The quantities tracked in this paper seem to be well-designed owing to a good abstraction of existing proof techniques.

**Weaknesses:**

- Several possible alternatives discussed by the authors for smooth non-convex optimization do not seem to contribute much to the "theoretical understanding of optimization algorithms used in deep learning": a) Weak convexity still might not hold globally; b) The algorithms for non-convex non-smooth optimization (e.g. [1, 2]) might be too complicated to be practical; c) The variants without random scaling of existing algorithms seem to perform on par with the variants with random scaling.

[1] Zhang, J., Lin, H., Jegelka, S., Sra, S. and Jadbabaie, A., 2020, November. Complexity of finding stationary points of nonconvex nonsmooth functions. In International Conference on Machine Learning (pp. 11173-11182). PMLR.

[2] Jordan, M., Kornowski, G., Lin, T., Shamir, O. and Zampetakis, M., 2023, July. Deterministic nonsmooth nonconvex optimization. In The Thirty Sixth Annual Conference on Learning Theory (pp. 4570-4597). PMLR.

**Questions:**

- Sections 3.1 and 3.2 appear to convey contradictory messages about GPT-2. The last plot in Figure 2 suggests that GPT-2 is locally convex, while the final plot in Figure 3 indicates a lack of convexity. Could you clarify this?
- Allen-Zhu et al. [3] proved that the ReLU networks are semi-smooth with high probability. Could you please comment on this? Do your experimental results violate their theoretical results?
- In the interpolation regime and x_t is close to y_t = x_*, both the numerator and denominator in (4) should converge to 0. Why did this quantity explode at the end in Figure 4?


[3] Allen-Zhu, Zeyuan, Yuanzhi Li, and Zhao Song. "A convergence theory for deep learning via over-parameterization." In International conference on machine learning, pp. 242-252. PMLR, 2019.

---

### Official Review · Reviewer_Nwr1 · 2024-11-04

**Soundness:** 3
**Presentation:** 3
**Contribution:** 3
**Rating:** 6
**Confidence:** 3

**Summary:**

This paper studies whether the standard bounds used to analyze optimization actually hold in deep learning settings. First, the paper shows that optimizers sometimes find "locally convex" paths, but argues that convexity-based analysis cannot explain optimization in deep learning. It then considers smoothness and observes that the smoothness is connected to the learning rate, similar to Cohen et al. 2020. Finally, they observe that update correlations are often positive during training, which challenges analyses that rely on monotonic loss decrease in expectation.

**Strengths:**

- The primary goal of the paper – verifying the inequalities used in optimization theory – is clear and well motivated.
- The paper is overall well written. The paper connects high level assumptions to specific inequalities, and each section describes how the corresponding inequality is used in standard analyses and how the paper will attempt to verify them.
- The overall results, while not entirely surprising, strongly support the claim that existing analyses are unable to explain convergence in realistic settings.

**Weaknesses:**

- [1] proposed "directional smoothness" which is the same as this paper's notion of instantaneous smoothness. They also observed that this quantity can approximate the sharpness.
- The experiments and conclusion in section 4.1 are essentially equivalent to those in Cohen et al. 2020;2022.
- While not exactly equal to the update correlation (eq. 9), Cohen et al. 2020 Appendix H showed that for SGD, the loss does not decrease in expectation which carries a similar message to section 4.2.
- Appendix D is a bit rough and many of the figures are missing discussions. It may also make sense to discuss some of these findings in the main paper, especially Appendix D.1.

Minor Comments
- lines 373-377: this can be easily explained by the observation in Cohen et al. 2020 that at the edge of stability, the optimizer oscillates in the top eigenvector direction, and if $x_t$ and $x_{t-1}$ differ in the top eigenvector direction then a first order Taylor expansion shows that the instantaneous smoothness is approximately the sharpness.

[1] Mishkin et al. 2024: Directional Smoothness and Gradient Methods: Convergence and Adaptivity

**Questions:**

- Could the authors clarify how the discussion in section 4 differs from the observations in Cohen et al. 2020?
- Could the authors elaborate on the motivation for the additional experiments in Appendix D, especially D.3, D.4, and D.8?
- Have the authors considered the effects of the reference point on these metrics? For example, is it possible that with a better reference point $\bar x$ (e.g. found by decaying the learning rate or averaging weights across an epoch) that the average update correlation $\langle \nabla f(\bar x), x_{t+1} - x_t \rangle$ would be consistently negative?

---

### Official Review · Reviewer_if1T · 2024-11-04

**Soundness:** 3
**Presentation:** 3
**Contribution:** 2
**Rating:** 5
**Confidence:** 3

**Summary:**

This paper revisits two commonly used assumptions (convexity and smoothness) for analyzing the convergence of optimization algorithms. Traditionally, theoretical understanding of optimization algorithms usually requires assumptions of global convexity and global smoothness. This paper numerically investigates these two assumptions by examining the optimization path for algorithms like GD/SGD/AdamW, and finds that in most cases the assumptions do not hold. The results suggest that new theoretical frameworks are needed.

**Strengths:**

1. The organization and presentation of this paper is smooth and clear, and it provides new insights when comparing modern training algorithms SGD and AdamW.

2. This paper illustrates the necessity for developing a new theoretical framework to theoretically analyze optimization algorithms, which is crucial for this deep learning era.

**Weaknesses:**

In case I miss something, please correct me if I am wrong.

My biggest concern is that the main findings in this paper is not that surprising, i.e., it is expected that in the practical training, the loss landscape is highly non-convex, and thus it is reasonable if the convexity or the smoothness does not hold along the whole optimization path. In a high level point of view, this paper seems to reveal the gap between theory and practice, but this gap has been around since the days of deep learning. I would expect the authors propose some reasonable assumptions to enhance the theoretical understanding of modern training algorithms.

**Questions:**

1. For the numerical experiments, since the initialization is randomly chosen, so how will the random initialization affects the numerical results?

2. How will the choice of learning rate affect the numerical results?

---

### Note · Authors · 2024-11-19

I have read and agree with the venue's withdrawal policy on behalf of myself and my co-authors.